# Evidence of *Histoplasma capsulatum* seropositivity and exploration of risk factors for exposure in Busia county, western Kenya: Analysis of the PAZ dataset

**Tessa Rose Cornell**[1]*, **Lian Francesca Thomas**[1,2], **Elizabeth Anne Jessie Cook**[2], **Gina Pinchbeck**[1], **Judy Bettridge**[1,2¤a¤b], **Lauren Gordon**[1¤c], **Velma Kivali**[2], **Alice Kiyong'a**[2], **Eric Maurice Fèvre**[1,2], **Claire Elizabeth Scantlebury**[1]

**1** Institute of Infection, Veterinary and Ecological Sciences (IVES), University of Liverpool, Liverpool, United Kingdom, **2** International Livestock Research Institute (ILRI), Nairobi, Kenya

¤a Current address: Natural Resources Institute, University of Greenwich, Chatham Maritime United Kingdom
¤b Current address: Bristol Veterinary School, University of Bristol, Bristol, United Kingdom
¤c Current address: Cellmark Forensic Services, Abingdon, United Kingdom
* Tessa.Cornell@liverpool.ac.uk

**Data Availability Statement:** The original dataset, and the serology results, are available via an open access repository held by the University of

## Abstract

### Background

Despite recognition of histoplasmosis as a disease of national public health concern in Kenya, the burden of *Histoplasma capsulatum* in the general population remains unknown. This study examined the human seroprevalence of anti-*Histoplasma* antibody and explored associations between seropositivity and demographic and environmental variables, in Busia county, western Kenya.

### Methodology

Biobanked serum samples and associated data, from a previous cross-sectional survey, were examined. Latex agglutination tests to detect the presence of anti-*Histoplasma* antibody were performed on serum samples from 670 survey respondents, representing 178 households within 102 sub-locations.

Potential epidemiologic risk factors for *H. capsulatum* exposure were explored using multi-level multivariable logistic regression analysis with household and sub-location included as random effects.

### Principal findings

The apparent sample seroprevalence of anti-*Histoplasma* antibody was 15.5% (*n* = 104/670, 95% Confidence Interval (CI) 12.9–18.5%). A multivariable logistic regression model identified increased odds of *H. capsulatum* seropositivity in respondents reporting rats within the household within the previous 12 months (OR = 2.99 90% CI 1.04–8.55, *p* = 0.04).

Liverpool (http://dx.doi.org/10.17638/datacat.liverpool.ac.uk/352).

**Funding:** The original samples described were collected with the support of the Wellcome Trust (085308 to EMF) and a Medical Research Council DTG award (G1000388 to EAJC). This work received support from the CGIAR Research Program on Agriculture for Nutrition and Health (A4NH; to EMF) and we thank the CGIAR Fund donors (https://www.cgiar.org/funders/). Laboratory processing and analyses presented here were supported by funds from a Wellcome Trust ISSF Fellowship award (204822/Z/16/Z to CES). The funders had no role in this study design, analysis, manuscript preparation or decision to publish.

**Competing interests:** The authors have declared that no competing interests exist.

Compared to respondents aged 25–34 years, the odds of seropositivity were higher in respondents aged 15–24 years (OR = 2.70 90% CI 1.04–6.97, $p$ = 0.04).

## Conclusions

The seroprevalence result provides a baseline for sample size approximations for future epidemiologic studies of the burden of *H. capsulatum* exposure in Busia county. The final model explored theoretically plausible risk factors for *H. capsulatum* exposure in the region. A number of factors may contribute to the complex epidemiological picture impacting *H. capsulatum* exposure status at the human-animal-environment interface in western Kenya. Focussed *H. capsulatum* research is warranted to determine the contextual significance of identified associations, and in representative sample populations.

## Author summary

Despite recognition of histoplasmosis as a priority disease of public health concern in Kenya and an important AIDS-defining illness, there remains a paucity of research on this neglected fungal disease. Clinical and laboratory capacity for the diagnosis and treatment of histoplasmosis across Kenya is limited or unknown, and existing diagnostic and therapeutic techniques can be cost-prohibitive. In addition, the fragmentary nature of histoplasmosis research groups worldwide and the under- or over-representation of specific sociodemographic groups and geographic regions in outbreak reports and hospital-based case series have been acknowledged.

This study provides a first look at *Histoplasma capsulatum* seroprevalence in rural western Kenya and explores risk factors for exposure at this human-animal-environment interface. More broadly, these outcomes will help quantify the burden of *H. capsulatum* in household and community environments, which may direct further research efforts and inform policy-makers on the prioritisation for clinical services and public health efforts with regards to histoplasmosis.

## Introduction

The burden of *Histoplasma capsulatum* is sparsely documented in sub-Saharan Africa, including in Kenya where histoplasmosis has been recognised as a priority disease of national public health concern [1,2]. Histoplasmin skin sensitivity surveys conducted in a limited number of countries in sub-Saharan Africa have recorded test positivity rates between 0.0 and 35.0% in populations with variable demographic and clinical characteristics [3–11]. These findings indicate that *H. capsulatum* is present within this geographic region. However, further research is warranted to explore the factors contributing to varying prevalence between different geographic areas and environments, the risk factors for exposure and infection, and the incidence and clinical outcomes of histoplasmosis.

The limited research in this area is confounded by multiple barriers to the identification and management of human histoplasmosis, which comprise: (i) case under-reporting; (ii) case mis-diagnosis; (iii) limited access to clinical facilities for case diagnosis or treatment; (iv) limited access to anti-fungal treatments; (v) cost-prohibitive diagnostic or treatment methods; and (vi) poor definition of transmission routes and risk factors for exposure [12–15]. These

barriers present a significant challenge to histoplasmosis surveillance, treatment, and infection control and thus limit our understanding of how *H. capsulatum* exposure or infection impacts the Kenyan population.

A number of risk factors for histoplasmosis are widely acknowledged; however, evidence of contextual factors relevant to sub-Saharan Africa remains limited. Disseminated histoplasmosis has been identified as a major AIDS-defining disease presentation of HIV-infected patients [16]. In contrast to disease course in immunocompetent hosts, which is typically characterised as asymptomatic and self-limiting [17], patients with *H. capsulatum* and HIV co-infection have demonstrated significant morbidity and mortality rates in the absence of appropriate treatment [18,19].

Occupational and recreational activities speculated to increase risk of aerosolisation and inhalation of infective *H. capsulatum* microconidia have been described in histoplasmosis case and outbreak reports and hospital-based case series. Tunnel work [20], land excavation [21], bat habitat exposure during cave and tunnel visits [22,23], and exposure to bird faeces and roosts [24,25] have been reported as plausible risk factors for *H. capsulatum* exposure. *H. capsulatum* has been identified in soil and water samples [26–28], bats [29], and bat and bird faeces [30–34] by direct microscopy, mouse inoculation and culture technique, or molecular detection. In the rural Kenyan context, humans can live in close proximity with domestic and wild animals, and previously recognised reservoirs of *Histoplasma* could be present within household environments.

The current study utilised serum samples and data on demographic and animal exposure variables previously collected during a cross-sectional household survey in Busia county, western Kenya [35] to explore levels of exposure to *H. capsulatum*. The primary objectives of the study were as follows:

- Estimate the human seroprevalence of anti-*Histoplasma* antibody in Busia county, using a latex agglutination test (LAT);

- Explore associations between *H. capsulatum* seropositivity, and non-clinical demographic and environmental variables in Busia county; and

- Identify limitations in current data with regards to identifying the burden of *H. capsulatum* exposure in the Kenyan context and thus highlight future research activities required to address gaps in evidence.

## Methods

### Ethics statement

Ethics approval for serum sample collection and storage for future processing was granted by the Kenya Medical Research Institute Ethics Review Committee (ERC; reference SSC/1701). Permission to re-analyse data and test bio-banked serum samples was provided by KEMRI and supported by Scantlebury Wellcome Trust ISSF Fellowship (reference 204822/Z/16/Z).

Written informed consent was obtained for all adults (>15 years) and assent was obtained on behalf of children (5–15 years) by their legal guardian. All survey respondents were ≥5 years.

### Original study

A cross-sectional household survey was conducted from 2010–12 in Busia county, western Kenya, for the People, Animals and their Zoonoses (PAZ) project, supported by the Wellcome Trust [36]. The study region was selected as it broadly represents the wider Lake Victoria

**Table 1. Selection of study respondents (n = 670/942, 71.1%) from the household survey (Fèvre et al., 2017 [35]), characterised by HIV status (Negative/ Positive), and bat observation around the home in previous 12 months (No/ No response/ Yes).**

| | | Bats observed around home in previous 12 months | | | Total |
|---|---|---|---|---|---|
| | | No | No response | Yes | |
| HIV result | Negative | 291 | 1 | 330 | 622 |
| | Positive | 30 | 0 | 18 | 48 |
| Total | | 321 | 1 | 348 | 670 |

Crescent ecosystem, namely that of a smallholder, mixed crop-livestock production system, with previously poorly understood burden of zoonotic infection [35]. Project outputs included epidemiologic data on the prevalence of neglected zoonotic diseases amongst 2113 survey respondents, from randomly selected households stratified by sub-location (*n* = 143) [35,37]. For complete methodology of household selection, refer to Fèvre *et al.* (2017) [35].

## Serum sample selection

A sub-set (*n* = 942) of samples was selected from the PAZ dataset and represents respondents who received a question on bat observation. The data sub-set was collected between May 2011 and July 2012. Within this time period, the following criteria were applied to select the study sample of 670 survey respondents (*n* = 670/942, 71.1%). Serum samples were selected to include survey respondents reporting variable HIV status at the time of sampling (positive: *n* = 48/670, 7.2%), and absence or presence of bats around the home during the 12 months prior to survey delivery (bats observed: *n* = 348/670, 51.9%) (Table 1). All respondents with a HIV positive status and all those reporting the presence of bats were selected, in addition to systematically selecting every eighth respondent across the dataset. Selected survey respondents represented 178 households within 102 sub-locations.

## Serological testing

One IMMY Latex Agglutination *Histoplasma* test was performed as per manufacturer guidelines for each thawed, heat-treated serum sample. In accordance with IMMY guidelines, a graduated scale of reaction strengths was used to assign test results from negative (-) to four plus (4+) (S1 Fig). Positive and negative controls had to demonstrate 2+ or greater, and less than 1+ reaction strengths, respectively [38]. Samples assigned a 2+ or greater reaction strength were considered to be presumptive evidence of active or recent *H. capsulatum* exposure. For the purpose of this study, no serum dilutions were performed as a measure of antibody titer.

The LAT provides a measure of agglutinating anti-*Histoplasma* antibody, predominant during the early IgM antibody response. Antibody responses in individuals with acute histoplasmosis have been characterised by an initial peak in IgM mean concentration at 14 to 27 days, before returning to pre-clinical levels by one year [39–41]. Thus, a positive LAT reaction can indicate acute infection from 2 weeks post-exposure. Furthermore, IgM levels have demonstrated a significant decrease between acute and convalescent phases 5–6 and 10–12 weeks post-exposure, respectively [42]. Implications of these test limitations for estimating true seroprevalence are discussed further.

## Seroprevalence estimation

The apparent prevalence of *H. capsulatum* seropositivity in the sample population was determined based on the IMMY Latex Agglutination *Histoplasma* test results. True seroprevalence

was estimated using published sensitivity and specificity values for a histoplasmin sensitised LAT of 62% and 97%, respectively [43]. Epitools interface and Clopper-Pearson (exact) test were employed to determine 95% Confidence Intervals (CIs) (https://epitools.ausvet.com.au/trueprevalence [44]).

## Statistical analysis

Household survey data and LAT results were stored in a protected Microsoft Excel file. Data variables encompassed non-clinical demographic and environmental factors and were selected for analysis if identified as either an established risk factor for *H. capsulatum* exposure in current literature, or a theoretically plausible risk factor for exposure based on current evidence of *H. capsulatum* life cycle and transmission dynamics (S1 and S2 Tables). Re-coding of variables is described in S2 Table.

Descriptive statistics were used to analyse respondent- and household level-characteristics of the selected sub-set of respondents and to compare this sub-set with the original sampled population. The Mann-Whitney U test was applied to compare distributions of categorical variables between the original sample and the sub-set and to determine the statistical significance of differences.

Univariable associations between *H. capsulatum* seropositivity and individual selected variables were examined by constructing 2xN contingency tables and using Pearson Chi-squared test of association ($\chi^2$). Odds Ratios (ORs) with 90% CIs and associated *p*-values were calculated.

Phi coefficient was employed to analyse suspected correlations between categorical variables with a binary outcome. A coefficient value of >0.7 with an associated *p*-value <0.05 was interpreted as evidence of an association between variables. Identification of an association and subsequent comparison of *p* values on univariable analysis supported exclusion of variables from further analysis, whereas those with stronger *p*-values were retained for further analysis.

Variables with a $\chi^2$-associated *p*-value <0.20 on univariable analysis were selected for testing in a multivariable logistic regression model with seropositivity as the binary outcome. The model was built using a manual backwards-stepwise approach [45]. As the study is exploratory and designed to generate hypotheses about potential risk factors for *H. capsulatum* exposure a conservative cut-off value of *p*<0.10 was applied to include variables in the final model. Final versions of the model were assessed using the Hosmer Lemeshow test statistic, and Delta Betas were explored for variables within the final model to examine the effect of any influential data points. Random effects were included to explore the effect of clustering of respondents at both household and sub-location levels. Regression coefficients, estimate *p*-values, and z-ratios were compared between single and multi-level models. Proportion of variance attributed to individual levels was calculated using the latent-variable approach described by Goldstein *et al.* (2002) [46].

Statistical analyses and multi-level modelling were performed using IBM SPSS Statistics 25, and MLwiN 3.05 software, respectively.

## Results

### Study population

Selected respondents (*n* = 670/942, 71.1%) represented 178 households, with a median number of occupants of 7.0 per household (range: 1–30). The sample comprised respondents aged 5 to >85 years and displayed a positively skewed age distribution (S2 Fig). Modal and median age categories were 5–14 years (*n* = 272/670, 40.6%), and 15–24 years (*n* = 133/670, 19.9%),

respectively. The sample comprised 350 females (52.2%). The gender ratio per age category was approximately 90–125 females per 100 males, with the exception of age category 25–34 years which demonstrated a greater gender gap of 213 females per 100 males. The majority of respondents were teachers or students (*n* = 341/670, 50.9%) or within animal management or contact roles (*n* = 244/670, 36.4%). A positive HIV status was recorded in 7.2% (*n* = 48/670) of respondents, with the highest prevalence of HIV positive status among respondents aged 35–44 years (*n* = 18/670, 23.7%), and females (Female: *n* = 33/350, 9.4%; Male: *n* = 15/320, 4.7%). A minority of respondents reported smoking behaviour (*n* = 17/670, 2.5%) (Tables 2 and S3).

Contact with dogs (*n* = 582/670, 86.9%), cats (*n* = 563/670, 84.0%), and poultry (*n* = 597/670, 89.1%), and observation of rats (*n* = 609/670, 90.9%) in and around the household environment were reported by the majority of respondents. Examination of indirect animal contact activities demonstrated that the majority of respondents were involved in manure preparation (*n* = 436/670, 65.1%), in contrast to animal burial and skinning activities which were not frequently reported (Table 2). The most common manure preparation activities were described as preparation for fuel and use as a building material.

At a household level (*n* = 178 households), spring water (wet: *n* = 93/178, 52.2%; dry: *n* = 92/178, 51.7%) and borehole sources (wet: *n* = 73/178, 41.0%; dry: *n* = 74/178, 41.6%) were most frequently reported in the previous wet and dry seasons. Houses within the study area tended to be constructed of: iron (*n* = 128/178, 71.9%) or thatch (*n* = 121/178, 68.0%) roofs; mud walls (*n* = 160/178, 89.9%); and earth floors (*n* = 156/178, 87.6%). The majority of households reported using firewood as the main cooking fuel (*n* = 136/178, 76.4%) (Table 2).

No statistically significant differences were found in the demographic and behavioural variables between the original sample and the sub-set of respondents selected for this study, with the exception of observation of bats around the home which was significantly higher in the sub-set (*p*<0.001) due to the selection criteria applied for sub-setting (S4 Table). We therefore consider the sub-set selected for this study to be appropriately representative of the underlying population, enabling population inferences to be made with regards to seroprevalence and risk factors being explored.

## Seroprevalence results

A total of 104 serum samples were interpreted as positive on LAT (a reaction strength of 2+ or greater), of which the majority displayed a reaction strength of 2+ (*n* = 68/104, 65.4%) (Fig 1 and Table 3). This related to an apparent seroprevalence of 15.5% (*n* = 104/670, 95% CI 12.9–18.5%). The estimated true seroprevalence in this sample, adjusting for published LAT sensitivity and specificity results [43], was calculated as 21.2% (95% CI 16.8–26.2%) [44].

43.3% of households (*n* = 77/178) contained at least one occupant with a seropositive result. Of these households, the percentage of occupants demonstrating seropositivity ranged from 7.7% (*n* = 1/13 occupants) to 100.0% (*n* = 1/1 to 3/3 occupants) (Fig 2A and 2B).

## Univariable analysis

Univariable logistic regression analysis identified a statistically significant association (*p*<0.05) between LAT result and age category 15–24 years (OR = 2.80 90% CI 1.28–6.15, *p* = 0.03; reference category 25–34 years). Variables which met the multivariable model inclusion criteria at a higher *p*-value cut-off (*p*<0.1) were as follows; observation of rats around the home in the previous 12 months (OR = 2.80 90% CI 1.17–6.68, *p* = 0.05), use of spring water in the previous dry season (OR = 1.47 90% CI 1.02–2.13, *p* = 0.08), and age category ≥45 years (OR = 2.41 90% CI 1.08–5.39, *p* = 0.07; reference category 25–34 years) (Table 2).

**Table 2. Univariable logistic regression analysis results, examining associations between *H. capsulatum* seropositivity based on LAT results, and respondent- and household-level variables, amongst survey respondents (*n* = 670/942) in Busia county, western Kenya.** Frequencies (n), percentages (%), Odds Ratios (OR), 90% Confidence Intervals (CIs) and *p*-values, were calculated using IBM SPSS Statistics 25 software.

| Variable | Frequency, n (%), total N = 670 | *H. capsulatum* seropositive, n (%), total N = 104 | *H. capsulatum* seronegative, n (%), total N = 566 | Odds Ratio (90% CI) | *p*-value |
|---|---|---|---|---|---|
| Demographic | | | | | |
| Sex | | | | | |
| Male | 320 (47.8) | 56 (17.5) | 264 (82.5) | 1.00 | |
| Female | 350 (52.2) | 48 (13.7) | 302 (86.3) | 0.75 (0.53–1.07) | 0.18 |
| Age category, years | | | | | |
| 5–14 | 272 (40.6) | 38 (14.0) | 234 (86.0) | 1.79 (0.84–3.81) | 0.21 |
| 15–24 | 133 (19.9) | 27 (20.3) | 106 (79.7) | 2.80 (1.28–6.15) | 0.03* |
| 25–34 | 72 (10.7) | 6 (8.3) | 66 (91.7) | 1.00 | |
| 35–44 | 76 (11.3) | 12 (15.8) | 64 (84.2) | 2.06 (0.86–4.93) | 0.17 |
| ≥45 | 117 (17.5) | 21 (17.9) | 96 (82.1) | 2.41 (1.08–5.39) | 0.07** |
| Occupation | | | | | |
| Animal contact roles | 244 (36.4) | 38 (15.6) | 206 (84.4) | 1.00 | |
| Not applicable or None (or no answer, *n* = 1) | 16 (2.4) | 1 (6.3) | 15 (93.8) | 0.36 (0.06–2.03) | 0.33 |
| Building roles | 16 (2.4) | 2 (12.5) | 14 (87.5) | 0.77 (0.22–2.78) | 0.74 |
| Teacher or Student | 341 (50.9) | 53 (15.5) | 288 (84.5) | 1.00 (0.68–1.46) | 0.99 |
| Trader | 28 (4.2) | 5 (17.9) | 23 (82.1) | 1.18 (0.50–2.79) | 0.75 |
| Other (S3 Table) | 25 (3.7) | 5 (20.0) | 20 (80.0) | 1.36 (0.57–3.24) | 0.57 |
| Clinical | | | | | |
| HIV status | | | | | |
| Negative | 622 (92.8) | 100 (16.1) | 522 (83.9) | 1.00 | |
| Positive | 48 (7.2) | 4 (8.3) | 44 (91.7) | 0.48 (0.20–1.14) | 0.16 |
| Smoking behaviour | | | | | |
| No | 653 (97.5) | 100 (15.3) | 553 (84.7) | 1.00 | |
| Yes | 17 (2.5) | 4 (23.5) | 13 (76.5) | 1.70 (0.65–4.43) | 0.36 |
| Domestic animal or livestock contact | | | | | |
| Dog contact | | | | | |
| No | 88 (13.1) | 12 (13.6) | 76 (86.4) | 1.00 | |
| Yes | 582 (86.9) | 92 (15.8) | 490 (84.2) | 1.19 (0.69–2.05) | 0.60 |
| Cat contact | | | | | |
| No | 107 (16.0) | 20 (18.7) | 87 (81.3) | 1.00 | |
| Yes | 563 (84.0) | 84 (14.9) | 479 (85.1) | 0.76 (0.49–1.20) | 0.32 |
| Poultry building access | | | | | |
| No | 73 (10.9) | 11 (15.1) | 62 (84.9) | 1.00 | |
| Yes | 597 (89.1) | 93 (15.6) | 504 (84.4) | 1.04 (0.59–1.84) | 0.91 |
| Pigs building access | | | | | |
| No | 642 (95.8) | 99 (15.4) | 543 (84.6) | 1.00 | |
| Yes | 28 (4.2) | 5 (17.9) | 23 (82.1) | 1.19 (0.44–3.21) | 0.73 |
| Shoats building access | | | | | |
| No | 639 (95.4) | 99 (15.5) | 540 (84.5) | 1.00 | |
| Yes | 31 (4.6) | 5 (16.1) | 26 (83.9) | 1.05 (0.46–2.39) | 0.92 |
| Cattle building access | | | | | |
| No | 637 (95.1) | 101 (15.9) | 536 (84.1) | 1.00 | |
| Yes | 33 (4.9) | 3 (9.1) | 30 (90.9) | 0.53 (0.19–1.46) | 0.30 |
| Burying dead animals | | | | | |

*(Continued)*

**Table 2.** (Continued)

| Variable | Frequency, n (%), total N = 670 | *H. capsulatum* seropositive, n (%), total N = 104 | *H. capsulatum* seronegative, n (%), total N = 566 | Odds Ratio (90% CI) | *p*-value |
|---|---|---|---|---|---|
| No | 658 (98.2) | 103 (15.7) | 555 (84.3) | 1.00 | |
| Yes | 12 (1.8) | 1 (8.3) | 11 (91.7) | 0.49 (0.09–2.76) | 0.50 |
| Skinning dead animals | | | | | |
| No | 656 (97.9) | 100 (15.2) | 556 (84.8) | 1.00 | |
| Yes | 14 (2.1) | 4 (28.6) | 10 (71.4) | 2.22 (0.83–5.98) | 0.18 |
| Manure preparation | | | | | |
| No | 234 (34.9) | 35 (15.0) | 199 (85.0) | 1.00 | |
| Yes | 436 (65.1) | 69 (15.8) | 367 (84.2) | 1.07 (0.74–1.55) | 0.77 |
| **Wildlife observation around home** | | | | | |
| Bats[1] | | | | | |
| No | 321 (48.0) | 43 (13.4) | 278 (86.6) | 1.00 | |
| Yes | 348 (52.0) | 61 (17.5) | 287 (82.5) | 1.37 (0.96–1.96) | 0.14 |
| Rats | | | | | |
| No | 61 (9.1) | 4 (6.6) | 57 (93.4) | 1.00 | |
| Yes | 609 (90.9) | 100 (16.4) | 509 (83.6) | 2.80 (1.17–6.68) | 0.05** |
| Wild birds[1] | | | | | |
| No | 554 (82.8) | 88 (15.9) | 466 (84.1) | 1.00 | |
| Yes | 115 (17.2) | 16 (13.9) | 99 (86.1) | 0.86 (0.53–1.39) | 0.60 |
| **Household building materials** | | | | | |
| Mud wall(s) | | | | | |
| No | 51 (7.6) | 4 (7.8) | 47 (92.2) | 1.00 | |
| Yes | 619 (92.4) | 100 (16.2) | 519 (83.8) | 2.26 (0.94–5.43) | 0.13 |
| Earth floor(s) | | | | | |
| No | 63 (9.4) | 6 (9.5) | 57 (90.5) | 1.00 | |
| Yes | 607 (90.6) | 98 (16.1) | 509 (83.9) | 1.83 (0.88–3.79) | 0.17 |
| Thatch roof(s) | | | | | |
| No | 189 (28.2) | 24 (12.7) | 165 (87.3) | 1.00 | |
| Yes | 481 (71.8) | 80 (16.6) | 401 (83.4) | 1.37 (0.91–2.07) | 0.21 |
| **Water source (last dry season)** | | | | | |
| Spring | | | | | |
| No | 284 (42.4) | 99 (34.9) | 185 (65.1) | 1.00 | |
| Yes | 386 (57.6) | 5 (1.3) | 381 (98.7) | 1.47 (1.02–2.13) | 0.08** |
| **Main cooking fuel in household** | | | | | |
| Main cooking fuel | | | | | |
| Firewood open fire | 510 (76.1) | 82 (16.1) | 428 (83.9) | 1.00 | |
| Charcoal open fire | 7 (1.0) | 0 (0.0) | 7 (100.0) | - | 0.99 |
| Firewood and charcoal open fire | 153 (22.8) | 22 (14.4) | 131 (85.6) | - | 0.99 |

\* *p*<0.05 (statistically significant); ** *p*<0.1

[1] *n* = 1 respondent did not respond to questions on bat and wild bird observation. Respondent demonstrated LAT seronegative status.

For each water source variable, a statistically significant association (phi >0.7, *p*<0.05) was measured between reporting the use of the water source in the last wet season and reporting use of the same water source in the last dry season. Thus, only water source variables for the last dry season, specifically tap, spring, well, river and borehole sources, were included in further analyses, and the seasonal element of the variable was excluded. Findings from univariable logistic regression analysis are available in Table 2.

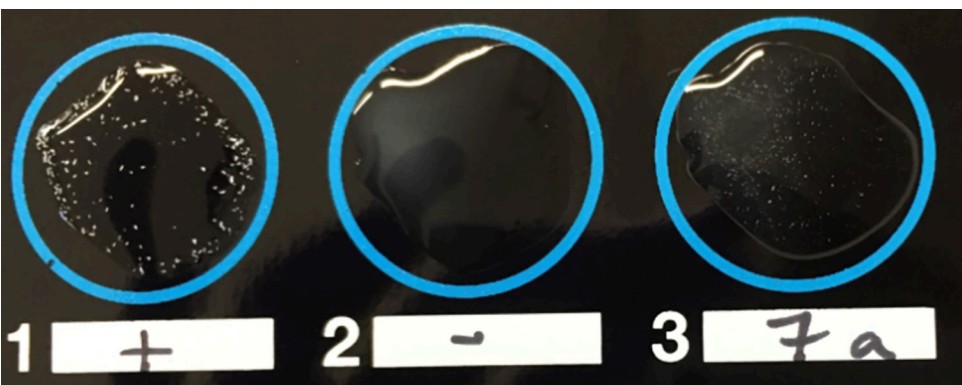

**Fig 1. IMMY Latex Agglutination-*Histoplasma* test demonstrating positive control (left), negative control (centre), and serum sample yielding positive result with a reaction strength of 2+ (right).**

## Multivariable logistic regression analysis

The final multi-level multivariable model contains two statistically significant main effects; observation of rats around the home in previous 12 months (OR = 2.99 90% CI 1.04–8.55, *p* = 0.04), and age category 15–24 (OR = 2.70 90% CI 1.04–6.97, *p* = 0.04; reference category 25–34 years), as variables associated with presence of anti-*Histoplasma* antibody (Table 4). The final model also included variables which met the inclusion criteria at a *p*-value cut-off of <0.1, as follows; age category ≥45 years (OR = 2.60 90% CI 0.98–6.89, *p* = 0.06; reference category 25–34 years) and mud walls in the household (OR = 2.50 90% CI 0.85–7.36, *p* = 0.097) (Table 4).

The model yielded a Hosmer-Lemeshow chi-squared value of 2.091 (*p* = 0.970). On delta beta analysis, no data points were determined to be influential on the model outcome.

Clustering by household (variance = 0.02, SE 0.18) and by sub-location (variance = 0.18, SE 0.17) were demonstrated. These outcomes indicated that only 0.6 and 5.2% of variance in seropositivity is due to household and sub-location respectively, using the latent-variable approach. Regression coefficients, z-ratios and *p*-values, of model variables in single-, two- and three-level models were comparable.

## Discussion

The study describes the human seroprevalence of anti-*Histoplasma* antibody, and explores associations between seropositivity and potential risk factors for *H. capsulatum* exposure in a community and household setting, in Busia county, western Kenya.

**Table 3. The frequency distribution of IMMY Latex Agglutination-*Histoplasma* test results for study respondents (*n* = 670/942), categorised by reaction strength, and result interpretation.**

|  |  | Description (IMMY, 2018 [38]) | Study respondents, *n* (%) |
|---|---|---|---|
| **LAT reaction strength** | - | A homogeneous suspension of particles with no visible clumping | 1 (0.1) |
|  | 1+ | Fine granulation against a milky background | 565 (84.3) |
|  | 2+ | Small but definite clumps against a slightly cloudy background | 68 (10.1) |
|  | 3+ | Large and small clumps against a clear background | 36 (5.4) |
|  | 4+ | Large clumps against a very clear background | 0 (0.0) |
| **LAT result interpretation** | Negative | Reaction strength–or 1+ | 566 (84.5) |
|  | Positive | Reaction strength 2+ to 4+ | 104 (15.5) |

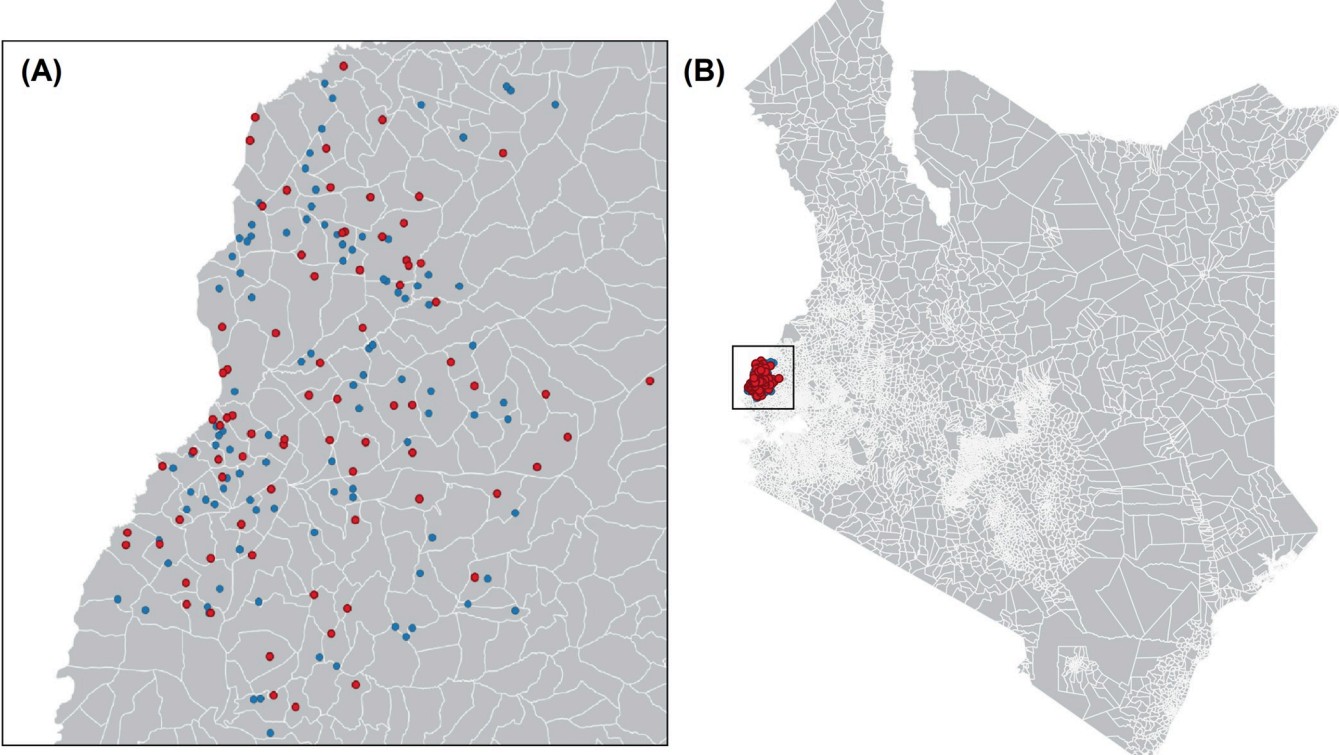

**Fig 2.** (A) Household locations (*n* = 178) within 102 sub-locations and (B) study area in Busia county, western Kenya. Key: red = households with ≥1 occupant seropositive for *H. capsulatum*; blue = households with zero seropositive occupants. Geodata were downloaded from DIVA-GIS (https://www.diva-gis.org/gdata). DIVA-GIS software is a free and open access source (https://www.diva-gis.org/docs/DIVA-GIS5_manual.pdf). Maps were generated using QGIS 3.12.

**Table 4. Multivariable logistic regression analysis examining variable associations with *H. capsulatum* seropositivity based on LAT results, amongst survey respondents (*n* = 670/942, 71.1%) in Busia county, western Kenya.** Odds Ratios (OR), 90% Confidence Intervals (CIs) and *p*-values, were calculated using MLwiN 3.05 software.

| Variable | *H. capsulatum* seropositive, *n* (%), N = 104 | *H. capsulatum* seronegative, *n* (%), N = 566 | Odds Ratio (90% Confidence Intervals) | *p*-value |
|---|---|---|---|---|
| Rats observed around home in previous 12 months | | | | |
| Reference category: No | 4 (6.6) | 57 (93.4) | 1.00 | |
| Yes | 100 (16.4) | 509 (83.6) | 2.99 (1.04–8.55) | 0.04* |
| Mud walls | | | | |
| Reference category: No | 4 (7.8) | 47 (92.2) | 1.00 | |
| Yes | 100 (16.2) | 519 (83.8) | 2.50 (0.85–7.36) | 0.097** |
| Age category, years | | | | |
| 5–14 | 38 (14.0) | 234 (86.0) | 1.72 (0.69–4.28) | 0.25 |
| 15–24 | 27 (20.3) | 106 (79.7) | 2.70 (1.04-'6.97) | 0.04* |
| Reference category: 25–34 | 6 (8.3) | 66 (91.7) | 1.00 | |
| 35–44 | 12 (15.8) | 64 (84.2) | 2.08 (0.72–5.97) | 0.17 |
| ≥45 | 21 (17.9) | 96 (82.1) | 2.60 (0.98–6.89) | 0.06** |

*\*p*-value <0.05 (statistically significant); \*\**p*-value <0.1

The recent recognition of histoplasmosis as a priority disease in Kenya [1] and apparent seroprevalence of *H. capsulatum* exposure demonstrated by survey respondents (*n* = 104/670, 15.5%), highlight the need for surveillance at national and regional levels. A previous histoplasmin skin test survey in Kenya reported a positivity rate of 8.5% (*n* = 65/768) in adult males [9]. The age distribution of the study population was not reported, and participants were miners or prisoners from Lake Victoria (western Kenya), and within or west of the Rift Valley, respectively. Skin test positivity is lower than our apparent measured seroprevalence of 15.5% (*n* = 104/670) which could be attributed to variable environmental conditions influencing survival of the saprophytic mycelial form of *H. capsulatum*, or variable exposure risk factors in the study populations under examination, including contact with animal reservoirs. In addition, the study described employs a histoplasmin skin sensitivity test as opposed to the LAT described in this study, which measure IgE-mediated reactions versus IgM agglutinating antibody responses, respectively.

With the exception of case reports [47–52], and limited prevalence studies in select sociodemographic groups [9,53,54], there is a paucity of recent epidemiologic data examining the burden of *H. capsulatum* exposure in the general population and in variable community and household settings in Kenya, the surrounding region, and more widely across sub-Saharan Africa. In Nigeria, two cross-sectional studies examining histoplasmin skin sensitivity across variable regions, demonstrated positive tests in 4.4% (*n* = 32/735) [8] and 10.5% (*n* = 69/660) [6] of participants. The latter study was conducted in proximity to a bat cave, and a sub-sample of this study population identified as farmers, cave guides and traders in the vicinity of the cave (35.0%, *n* = 14/40). Thus, the higher overall test positivity measured could be attributed to these study design factors.

Variables tested in the univariable and multivariable logistic regression models encompassed both established and theoretically plausible epidemiologic risk factors for *H. capsulatum* exposure. A significant association was identified between *H. capsulatum* seropositivity and the observation of rats within the household (OR = 2.99 90% CI 1.04–8.55, *p* = 0.04). In Kenya, *H. capsulatum* has been isolated from soil, including samples enriched with chicken and bat faeces [55–57]. Although evidence exists for the role of rats as environmental reservoirs, current literature is limited to North America, where *H. capsulatum* was identified in wild rats and soil samples proximal to rat burrows [26,58,59]. Additional research is warranted in the community setting in western Kenya to explore any associations between *H. capsulatum* exposure and the following variables: frequency and routes of human exposure to rats and their habitats, the location of rat burrows, isolation of *H. capsulatum* from rats and rat burrows, and the household and environmental factors maintaining rat populations.

The multivariable model presents an association (*p*<0.1) between *H. capsulatum* seropositivity and housing constructed with mud. Exploration of potential associations between building materials and the isolation of *H. capsulatum* in the household environment is warranted. One might hypothesise that mud walls may provide better substrate to maintain the saprophytic mycelial form of *H. capsulatum* in comparison to brick or cement. Furthermore, different building construction methods might present variable *H. capsulatum* exposure risks, for example construction of mud walls with handheld tools might increase exposure risk from soil. The variables presented may also be proxy indicators of socioeconomic factors that increase risk of *H. capsulatum* exposure, and could be indicative of the sociodemographic differences between regions, and availability of building materials. As stated, due to the exploratory nature of the study a conservative cut-off *p*-value was applied for inclusion of variables in the final model. Thus, strong conclusions should not be made based solely on these analyses and identified associations can contribute to generating hypotheses for future research.

Bat habitat exposure has been reported as a risk factor for *H. capsulatum* exposure [22,23], and *H. capsulatum* has been isolated from bats using molecular techniques [29]. The variable describing bat observation was not included in the final multivariable model. However, the purposeful selection of respondents reporting observation of bats around the home should be considered, which might increase overall seropositivity compared to a randomly selected sample. The difference between distributions of respondents reporting observation of bats in the study sample and in the original sub-set of respondents was statistically significant. Further investigation is warranted to examine the role of rats, bats and environmental reservoirs of *H. capsulatum* within this context. In addition, studies employing molecular methods may support current literature on phylogenetic characterisation of *Histoplasma* isolates and comparison to regional and global isolates from human, animal and environmental sources [60–62].

The multivariable logistic regression model demonstrates increased odds of seropositivity amongst age category 15–24 years (OR = 2.70 90% CI 1.04–6.97, $p = 0.04$) and an association ($p<0.1$) between seropositivity and age category ≥45 years (OR = 2.60 90% CI 0.98–6.89, $p = 0.06$), in comparison to respondents aged 25–34 years. Investigation of whether the outcome reflects variable immunocompetence between age categories, or age-related exposure to potential risk factors, is warranted. The sub-set of household survey respondents under examination demonstrated a positively skewed age distribution. At the time of data collection, in 2010, the age group 0–14 years represented 43.4% of the general Kenyan population [63]. This proportion is comparable to that of the sample population, of which 40.6% of selected respondents were 5–14 years ($n = 272/670$). Although representative of the general population, the effect of a skewed population structure on the frequency distribution of other variables under investigation, including reported occupations and involvement in animal contact roles, should be considered. For example, these variables may not be sufficiently powered to explore risk factors in older age categories. In comparison to studies exploring demographic or clinical risk factors for *H. capsulatum* infection in susceptible patient cohorts, the current study highlights potential environmental risk factors amongst the general population which may be confounded to a lesser extent by age.

Further targeted research is warranted to explore the impact of potential confounders such as age, gender and occupation. Investigation of associations between dwelling maintenance activities, building materials including mud walls, and presence of wild or domestic animals in occupied dwellings, would provide further objective insight into the interactions of described household and environmental variables, and their impact on *H. capsulatum* exposure risk.

Although the described associations do not infer direct causality, nor encompass the unknown lifestyle and socioeconomic confounding factors, the variables presented contribute to the complex epidemiological picture influencing *H. capsulatum* exposure status at the human-animal-environment interface in western Kenya.

There was no evidence for significant clustering at household- nor sub-location levels, however further investigation is warranted to identify the potential socio-demographic and geoclimatic variations between defined areas that have not been explored in this analysis and to quantify their impact on the odds of seropositivity.

*H. capsulatum* was not a focus of the original PAZ study [35], thus questions posed by the survey were not designed to capture risk factors relating specifically to *H. capsulatum* exposure nor to capture temporal information which might be related to the timing of exposure. Factors that contribute to whether inhalation of *H. capsulatum* microconidia results in symptomatic disease include the quantity of airborne inoculum and the immunocompetence of the host. A robust T cell response and subsequent activation of macrophages can prevent progression of *H. capsulatum* infection [64], in contrast to the progressive nature of infection in

immunocompromised individuals [65–67], however it should be noted that infection can become clinically apparent many years after first exposure.

With the exception of data on respondent smoking behaviour (yes: $n = 17/670$, 2.5% [seropositive: $n = 4/17$, 23.5%]) and HIV status (positive: $n = 48/670$, 7.2% [seropositive: $n = 4$, 8.3%]), clinical variables were excluded from analyses. The cross-sectional nature of data collection and absence of associated temporal data meant it was not possible to examine associations between reported clinical symptoms or disease and *H. capsulatum* seropositivity. Implementation of prospective, longitudinal research in community and household settings would enable more accurate inferences to be made about associations between *H. capsulatum* seropositivity and clinical signs or co-infections in the general population.

Hospital-based case series in Central and South America have examined morbidity and mortality in HIV-positive patients with confirmed disseminated histoplasmosis [68–70], and histoplasmosis is now widely recognised as a leading co-morbidity amongst AIDS patients [16]. An overall HIV prevalence of 7.7% of the general adult population was reported in Busia county in 2018 [71]. These individuals represent a potentially susceptible sub-set of the population to *H. capsulatum* co-infection. Clinical data with regards to individual immunocompetence of HIV-positive respondents at the time of survey delivery was not available, including access to and management of antiretroviral therapy. Among selected respondents, 48 (7.2%) demonstrated positive HIV status; however, no statistically significant association was identified between HIV positive status and *H. capsulatum* seropositivity. An examination of the impact of HIV infection on immunodiffusion and complement fixation test results, revealed detection of anti-*Histoplasma* antibodies was significantly lower ($p<0.05$) in disseminated histoplasmosis cases with, as opposed to without, AIDS [72]. Thus, we speculate that measured seropositivity amongst HIV positive respondents in this study could be an underestimate due to the inability of these individuals to mount an immune response detectable by LAT and subsequent false negative results. The effect of HIV co-infection on anti-*Histoplasma* antibody detection by LAT should be examined and quantified to improve our understanding of test performance and limitations and to increase the accuracy of seroprevalence estimates made on the basis of these test results. In addition, false positive results have been reported among patients with tuberculosis [73].

The possibility of cross-reactions with other systemic mycoses namely, *Aspergillus*, *Candida* and *Paracoccidioides* [74] should be acknowledged with the use of this LAT test. In the Kenyan context, *Aspergillus flavus* is documented as a major contaminant of maize crops, resulting in significant aflatoxin exposure [75]. Thus, the potential for cross-reactions with *Aspergillus*, specifically in a rural setting and in a maize-producing region, should be considered.

The LAT provides a measure of the presence of anti-*Histoplasma* antibody [38]. Only one LAT was performed per serum sample for the purpose of this study. Serum dilutions could also be performed as a semi-quantitative measure of antibody titer. The IMMY LA-*Histoplasma* test [38] references previously published overall sensitivity and specificity values [43]. However, test sensitivity ranged from 45.7 to 100%, for cases of chronic and acute primary pulmonary histoplasmosis, respectively [43]. Thus, the estimated true seroprevalence measured in the current study might vary significantly from 12.9% (95% CI 10.2–16.0%) to 29.3% (95% CI 23.2–36.2%) [44]. As the IgM antibody response to *Histoplasma* is mounted in 2–6 weeks, the LAT may present false negative results in individuals tested prior to 2 weeks post-exposure and after the IgM antibody response has diminished [39,41,42]. The samples were maintained temporarily at -20 degrees Celsius, prior to long term storage at -80 degrees Celsius. Freeze-thaw cycles were minimised and samples have not undergone any freeze-thaw cycles since 2016, thereby maintaining the integrity of samples for serological testing.

## Conclusions

Results from the current study suggest that exposure to *H. capsulatum* occurs frequently within this population and promotes the need for further longitudinal research to investigate the incidence of *H. capsulatum* exposure and infection in Kenya. The seroprevalence reported here may provide a baseline for sample size approximations to support future epidemiologic studies of the burden of histoplasmosis.

Exploration of theoretically plausible risk factors has highlighted areas for further investigation. Future research might focus on further examination of the significance of associations identified here, and consider how health, demographic, and socio-economic factors impact on *H. capsulatum* transmission at the human-animal-environment interface.

## Supporting information

**S1 Table. Variables selected for data analysis, from original survey human and household survey reports.** Analysis of observations of bats and wild birds only.
(DOCX)

**S2 Table. Original survey human and homestead report text, and re-coding of selected variables.** NR = not recorded; ND = not determined; NA = not applicable.
(DOCX)

**S3 Table. Re-categorised occupations for statistical analysis.**
(DOCX)

**S4 Table. Baseline characteristics of original sample ($n$ = 942) and selected sample ($n$ = 670) of survey respondents, and comparison of variable distribution differences using the Mann-Whitney U test.** * $p < 0.05$.
(DOCX)

**S1 Fig. IMMY Latex Agglutination-*Histoplasma* test reference images from equid serum demonstrating positive and negative controls (top row), and reaction strength grades negative and 1+ to 4+ (bottom row, left to right).**
(TIFF)

**S2 Fig. Histogram demonstrating the frequency distribution of age categories (years) for study respondents.**
(TIFF)

## Acknowledgments

We thank the PAZ field and laboratory teams in Busia and Nairobi, Kenya, for their work in collecting the samples and data under analysis.

## Author Contributions

**Conceptualization:** Tessa Rose Cornell, Claire Elizabeth Scantlebury.

**Data curation:** Tessa Rose Cornell, Lian Francesca Thomas, Elizabeth Anne Jessie Cook, Judy Bettridge, Lauren Gordon, Eric Maurice Fèvre, Claire Elizabeth Scantlebury.

**Formal analysis:** Tessa Rose Cornell, Gina Pinchbeck, Claire Elizabeth Scantlebury.

**Funding acquisition:** Eric Maurice Fèvre, Claire Elizabeth Scantlebury.

 _Histoplasma capsulatum_ seroprevalence and risk factors in western Kenya

**Investigation:** Tessa Rose Cornell, Judy Bettridge, Lauren Gordon, Velma Kivali, Alice Kiyong'a, Claire Elizabeth Scantlebury.

**Methodology:** Tessa Rose Cornell, Claire Elizabeth Scantlebury.

**Project administration:** Tessa Rose Cornell, Claire Elizabeth Scantlebury.

**Resources:** Eric Maurice Fèvre.

**Supervision:** Gina Pinchbeck, Claire Elizabeth Scantlebury.

**Validation:** Tessa Rose Cornell, Lian Francesca Thomas, Gina Pinchbeck, Claire Elizabeth Scantlebury.

**Visualization:** Tessa Rose Cornell.

**Writing – original draft:** Tessa Rose Cornell.

**Writing – review & editing:** Tessa Rose Cornell, Lian Francesca Thomas, Elizabeth Anne Jessie Cook, Gina Pinchbeck, Judy Bettridge, Lauren Gordon, Eric Maurice Fèvre, Claire Elizabeth Scantlebury.

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
