## [Decision Letter · Decision Letter 0]

26 Jan 2023

Dear Dr Cornell,

Thank you very much for submitting your manuscript "Evidence of Histoplasma capsulatum seropositivity and exploration of risk factors for exposure in Busia county, western Kenya: Analysis of the PAZ dataset" for consideration at PLOS Neglected Tropical Diseases. As with all papers reviewed by the journal, your manuscript was reviewed by members of the editorial board and by several independent reviewers. In light of the reviews (below this email), we would like to invite the resubmission of a significantly-revised version that takes into account the reviewers' comments. 

We cannot make any decision about publication until we have seen the revised manuscript and your response to the reviewers' comments. Your revised manuscript is also likely to be sent to reviewers for further evaluation.

Sincerely,

Joshua Nosanchuk, MD

Section Editor

Reviewer's Responses to Questions

**Key Review Criteria Required for Acceptance?**

**Methods**

-Are the objectives of the study clearly articulated with a clear testable hypothesis stated?

-Is the study design appropriate to address the stated objectives?

-Is the population clearly described and appropriate for the hypothesis being tested?

-Is the sample size sufficient to ensure adequate power to address the hypothesis being tested?

-Were correct statistical analysis used to support conclusions?

-Are there concerns about ethical or regulatory requirements being met?

Reviewer #1: The objectives of the study were clearly articulated. The study design was appropriate to address the stated objectives. The studied population was clearly described and appropriate for the hypothesis tested.

Regarding the sample size, I think that authors should describe the number of inhabitants in Busia county to really know how large is the size of population studied. In addition, authors should better explain why the survey respondents were selected from a sub-set of the PAZ dataset, representing respondents with available data on bat observation, which were 670/2113. Because I think the study is biased, and the seroprevalence figures are based only in this group, which is not representative of Busia.

Reviewer #2: Objectives are reasonably clearly stated. However, the text needs to make crystal clear with no ambiguity that the analysis in this paper explored associations of Histoplasma seropositivity and primarily non-clinical, demographic and environmental variables, so that no clinical conclusions should be drawn. The study design is appropriate for the purpose and the study is well carried out with a reasonable sample size. As expected with any epidemiological study, the analysis and the presentation thereof in the manuscript is statistics heavy. I request the authors to ensure that statistical techniques have been properly employed in this paper. I have some questions which I have noted in the annotated PDF.

**Results**

-Does the analysis presented match the analysis plan?

-Are the results clearly and completely presented?

-Are the figures (Tables, Images) of sufficient quality for clarity?

Reviewer #1: The analysis presented match the analysis plan. Results are clearly and completed presented. Figures and tables are ok.

Reviewer #2: The analytical approach is acceptable, and the results are reasonably well presented. However, there are two tables in the Supplementary Table list, which I think merit inclusion in the main manuscript. (I have more comments on this in the PDF annotations.) I would also like to see LAT images in which the + to +++ outcomes are clearly visible along with the controls for comparison. I think the composite LAT photo merits inclusion in the main manuscript.

**Conclusions**

-Are the conclusions supported by the data presented?

-Are the limitations of analysis clearly described?

-Do the authors discuss how these data can be helpful to advance our understanding of the topic under study?

-Is public health relevance addressed?

Reviewer #1: The conclusions are well presented.

Reviewer #2: The authors have done a reasonably good job of presenting conclusions based on their research results. However, I am of the opinion that the authors need to revisit and clarify their statistical analyses (which underpin the conclusions), so that strong conclusions are not drawn based on weak associations. The authors have invested a significant amount of words in the Discussion in order to describe limitations as well as the utility of the study, future studies, impact on individual and public health.

**Editorial and Data Presentation Modifications?**

Reviewer #1: (No Response)

Reviewer #2: (No Response)

**Summary and General Comments**

Reviewer #1: This study provides a first look at H. capsulatum seroprevalence in rural western Kenya and explores risk factors at this human-animal environment interface.

The study is well written and it is relevant. However, regarding rats, authors should emphasize that rats as well as dogs, are also an indicator of the presence of this pathogen in a given area. It should be pointed out that H. capsulatum infects numerous mammalian hosts, including humans. The infection is also very common in wild (e.g., marsupials, rodents, armadillos, lamas, sea mammals, sloths, bats) and domestic animals (e.g., dogs and cats) in endemic areas (Emmons 1950, Seyedmousavi et al. 2018). However, mammals appear to be dead-end hosts of Histoplasma since there is no person-to-person or animal-to-person spread. Bats have been proposed as a vector of spread, both because the fungus grows well in soil contaminated with bat guano and because bats themselves can be colonized with this fungus (Hoff & Bigler 1981). Regarding the sample size, I think that authors should describe the number of inhabitants in Busia county to really know how large is the size of population studied. In addition, authors should better explain why the survey respondents were selected from a sub-set of the PAZ dataset, representing respondents with available data on bat observation, which were 670/2113. Because I think the study is biased, and the seroprevalence figures are based only in this group, which is not representative of Busia.

Authors should also emphasized that seroprevalence might bi higher since the sensitivity of the test among HIV individuals is even lower than that described by the manufacturer or other researchers. I dont know if people receiving treatment with other immnunosuppressants such as corticosteroids for instance were included in this study. 

Despite these observation, the study is really very interesting, and it was well performed.

Reviewer #2: I have included my comments in the annotated PDF. You have done an excellent job with this study, but the manuscript requires a bit more spit and polish. Please pay attention to all parts of the manuscript including the references when revising.

PLOS authors have the option to publish the peer review history of their article (what does this mean?). If published, this will include your full peer review and any attached files.

Reviewer #1: No

Reviewer #2: No
---

## [Decision Letter · Decision Letter 1]

7 Apr 2023

Dear Dr Cornell,

We are pleased to inform you that your manuscript 'Evidence of Histoplasma capsulatum seropositivity and exploration of risk factors for exposure in Busia county, western Kenya: Analysis of the PAZ dataset' has been provisionally accepted for publication in PLOS Neglected Tropical Diseases.

Best regards,

Joshua Nosanchuk, MD

Section Editor

Joshua Nosanchuk

Section Editor

Reviewer's Responses to Questions

**Key Review Criteria Required for Acceptance?**

**Methods**

-Are the objectives of the study clearly articulated with a clear testable hypothesis stated?

-Is the study design appropriate to address the stated objectives?

-Is the population clearly described and appropriate for the hypothesis being tested?

-Is the sample size sufficient to ensure adequate power to address the hypothesis being tested?

-Were correct statistical analysis used to support conclusions?

-Are there concerns about ethical or regulatory requirements being met?

Reviewer #1: Objectives are clearly articulated, the study design is appropriated, statistical analysis is correct, the studied population is appropriate for the stated objectives. No ethical concerns were met.

Reviewer #2: The authors' modifications to all aspects of the manuscript were appropriate and acceptable. This is a well written article that would add value to the existing epidemiological literature on Histoplasmosis.

**Results**

-Does the analysis presented match the analysis plan?

-Are the results clearly and completely presented?

-Are the figures (Tables, Images) of sufficient quality for clarity?

Reviewer #1: Results are well-presented.

Reviewer #2: The authors' modifications to all aspects of the manuscript were appropriate and acceptable. The images look clear enough in the PDF of the revised manuscript.

**Conclusions**

-Are the conclusions supported by the data presented?

-Are the limitations of analysis clearly described?

-Do the authors discuss how these data can be helpful to advance our understanding of the topic under study?

-Is public health relevance addressed?

Reviewer #1: Conclusions are well-written.

Reviewer #2: The authors' modifications to all aspects of the manuscript were appropriate and acceptable. The changes have enhanced the value of the manuscript.

**Editorial and Data Presentation Modifications?**

Reviewer #1: I think the manuscript improved with all the suggested modifications carried out by the authors.

Reviewer #2: (No Response)

**Summary and General Comments**

Reviewer #1: This is an interesting work where authors reported the human seroprevalence of anti-Histoplasma antibody and explored associations between seropositivity and demographic and environmental variables, in Busia county, western Kenya.

Reviewer #2: (No Response)

PLOS authors have the option to publish the peer review history of their article (what does this mean?). If published, this will include your full peer review and any attached files.

Reviewer #1: No

Reviewer #2: No

---

## [Editor Report · Acceptance letter]

28 Apr 2023

Dear Dr Cornell,

We are delighted to inform you that your manuscript, "Evidence of *Histoplasma capsulatum* seropositivity and exploration of risk factors for exposure in Busia county, western Kenya: Analysis of the PAZ dataset," has been formally accepted for publication in PLOS Neglected Tropical Diseases.

Best regards,

Shaden Kamhawi

co-Editor-in-Chief

Paul Brindley

co-Editor-in-Chief
